# Molecular Design-Based Breeding: A Kinship Index-Based Selection Method for Complex Traits in Small Livestock Populations

**DOI:** 10.3390/genes14040807

**Published:** 2023-03-27

**Authors:** Jiamin Gu, Jianwei Guo, Zhenyang Zhang, Yuejin Xu, Qamar Raza Qadri, Zhe Zhang, Zhen Wang, Qishan Wang, Yuchun Pan

**Affiliations:** 1College of Animal Sciences, Zhejiang University, Hangzhou 310058, China; 2Key Laboratory of Livestock and Poultry Resources Evaluation and Utilization, Ministry of Agriculture and Rural Affairs, Hangzhou 310058, China; 3Zhejiang Key Laboratory of Dairy Cattle Genetic Improvement and Milk Quality Research, Hangzhou 310058, China; 4Hainan Institute, Zhejiang University, Building 11, Yongyou Industrial Park, Yazhou Bay Science and Technology City, Yazhou District, Sanya 572025, China; 5Hainan Yazhou Bay Seed Laboratory, Yongyou Industrial Park, Yazhou Bay Sci-Tech City, Sanya 572025, China; 6Department of Animal Science, School of Agriculture and Biology, Shanghai Jiao Tong University, Shanghai 200240, China

**Keywords:** animal breeding, breeding approach, KIS, indigenous pig breeds

## Abstract

Genomic selection (GS) techniques have improved animal breeding by enhancing the prediction accuracy of breeding values, particularly for traits that are difficult to measure and have low heritability, as well as reducing generation intervals. However, the requirement to establish genetic reference populations can limit the application of GS in pig breeds with small populations, especially when small populations make up most of the pig breeds worldwide. We aimed to propose a kinship index based selection (KIS) method, which defines an ideal individual with information on the beneficial genotypes for the target trait. Herein, the metric for assessing selection decisions is a beneficial genotypic similarity between the candidate and the ideal individual; thus, the KIS method can overcome the need for establishing genetic reference groups and continuous phenotype determination. We also performed a robustness test to make the method more aligned with reality. Simulation results revealed that compared to conventional genomic selection methods, the KIS method is feasible, particularly, when the population size is relatively small.

## 1. Introduction

China is home to approximately 84 indigenous pig breeds, which are rich in genetic resources and possess a wide range of desirable and undesirable traits. Some of the desirable traits include high meat quality, large litter size, high tolerance to roughage, and disease resistance, while undesirable traits include slow growth, high feed conversion ratio, and low lean meat rate. Examples of typical indigenous pig breeds include Jinhua and Laiwu pigs. To achieve pigs with desirable traits, one approach is to select indigenous breeds with superior performance or to create new breeds (lines) through crossbreeding between Chinese indigenous breeds and intensive breeds such as Duroc, Landrace, and Large White.

Before the widespread availability of high-throughput sequencing technology, best linear unbiased prediction (BLUP) was the most commonly used selection method in China, which is based on pedigree information and uses a kinship matrix to estimate breeding values. Despite its widespread use in early animal breeding, the accuracy of BLUP is limited by the quality of the pedigree information and performance test scale [1].

The utilization of genomic prediction, through the correlation of whole-genome genotypes with phenotypes in a large sample of individuals, has been established as a means of enhancing quantitative traits in domesticated animals [2,3,4]. However, the reliance on imputed genomic regions can result in decreased prediction accuracy when applied to populations with divergent linkage disequilibrium structures [5].

The advancement of next-generation sequencing technology has greatly increased the volume of pig genomic data, making it possible to utilize this information in breeding studies. The most commonly used method is genomic best linear unbiased prediction (GBLUP), which allows for the direct estimation of an individual’s genetic merit through the construction of a genomic relationship matrix G, instead of a kinship matrix A based on pedigree information. While GBLUP methods assume that all markers have the same genetic variance. In reality, only a few single nucleotide polymorphisms (SNPs) have an impact and are linked to the quantitative trait loci (QTL) affecting the trait, while most SNPs are ineffective [6,7,8]. 

In the realm of genetic prediction, Bayesian techniques have gained prominence due to their incorporation of marker effect variance as a predetermined prior distribution. This approach encompasses a range of variations, such as BayesA, BayesB, BayesC, Bayesian Lasso, among others. These methods address limitations in the single-trait genomic best linear unbiased prediction (GBLUP) method by assuming that only a few single nucleotide polymorphisms (SNPs) have significant effects, with the variance of these effects following an inverse chi-squared distribution. This assumption aligns more closely with reality, as only a few SNPs are usually linked to the quantitative trait loci (QTLs) affecting the trait, while most SNPs have no effect. As a result, the BayesB method is often more accurate when few QTLs control genetic variation with large effects [9,10,11].

Compared to GBLUP methods, Bayesian models have more estimated parameters, leading to increased computational effort and improved prediction accuracy. However, the computational speed and robustness are not as favorable as those of GBLUP [12,13]. Both Bayesian and GBLUP methods rely on the accuracy of phenotypic determination and require the creation of a large-scale reference population to ensure selection accuracy [8,9].

In the case of Chinese pig breeds with small foundation populations and the breeding work being carried out involves both intraspecific selection and the hybridization of locally adapted varieties with introduced varieties to create new hybrid varieties, a genetic reference population may not exist, and the determination of quantitative traits may not be precise due to a lack of highly qualified breeding staff and refined breeding equipment.

The main purpose of genomic selection is to estimate genome-wide SNP effects based on a reference population and then to sum the genetic effects of candidate individuals based on the specific alleles they carry. This means first identifying an ideal individual; then the more similar a candidate is to an ideal individual, the higher its breeding value. For this purpose, we propose a kinship index based selection (KIS) method. To validate the efficacy and stability of the KIS method, we performed a series of simulations using the MoBPS R package [14].

## 2. Materials and Methods

### 2.1. Methods

In the KIS method, the first step is to identify parental varieties that support the breeding objectives. The second step is to design breeding strategies; common breeding strategies include introgression cross breeding, stepwise cross breeding, and multiple cross breeding. The third step is to design ideal individuals with the desired genotype harboring parental advantageous traits, compare the differences between candidate and ideal individuals, and finally, select and cross the individuals with high similarity with the ideal individual. The visual representation of the process is shown in Figure 1A.

The establishment of the ideal individual is the core component of the KIS method, and the accuracy of this method is entirely dependent on the quality of the ideal individual. The first step in building the ideal individual is to identify the advantageous traits in the parent breed. Different pig breeds have specific advantages; the advantageous traits of Chinese indigenous pigs are mainly in meat quality, roughage tolerance, and disease resistance, while that of the introduced breeds has better leanness, growth rate, and feed conversion ratio [15]. The beneficial genotype corresponding to the desired trait is identified after the determination of the advantageous trait. The beneficial genotype can be established by GWAS, QTL database, pig resource census data, etc. The ideal individual containing the desired traits is established. In the actual operation process, the genome is divided into several subsystems based on traits or trait categories. The KIS method calculates the similarity between candidate individuals and ideal individuals based on each trait or trait category, and then synthesizes a comprehensive index (kinship index KI) by weighting according to importance. Considering the existence of multiple effects for one cause, we analyzed the interactions between genes and selected loci for consideration based on the effect value; loci with small effect values will be ignored. The process of selection is illustrated in Figure 1B. 

Based on the positional information of the ideal individual, the genotypes of the candidate individuals at these loci were extracted and compared to that of the ideal individual. The results of the comparison are expressed in the form of ordinals of 0, 1, and 2, where 0 represents both disadvantage alleles, 1 represents allelic heterozygosity with half advantageous alleles, and 2 represents advantageous alleles. The sum of the scores for all loci represents the degree of similarity between the candidate individual and the ideal individual. Candidate individuals are ranked according to their similarities to ideal individuals, and those with high similarities will be selected.

### 2.2. Simulation

#### 2.2.1. Historical Population

The historical population consisted of 200 animals, with an equal number of males and females. For each individual, an underlying true genetic value is calculated for each trait. Based on this, phenotypes can be generated [14]. Two quantitative traits containing 100 additive QTLs and 500 additive QTLs, respectively, were assigned to the historical population where each individual carries a varying number of QTLs. The genomic variance of these traits was simulated to be 1. According to the KIS method, two purebred populations, each to be selected for one quantitative trait, were created from the historical population. The purebred populations were raised separately for 10 generations. In each generation, 5 sires and 20 dams were crossed to obtain 200 offspring, of which the numbers of males and females were equal. The selection of sires and dams depended on the advantage of genotypic similarity with the ideal individual and was independent for the two purebred populations. Hereon, the two purebred populations are referred to as population A and population B. The simulation process for historical populations is shown in Figure 2A. Figure 2B shows principal components analysis of individuals in the founder population and generation 10 of population A and population B; the x-axis denotes the first principal component and the y-axis represents the second principal component.

#### 2.2.2. Breeding Program

We implemented hybridization and pure breeding programs to test the effectiveness of different breeding approaches. During the hybridization process, the first hybrid generation was obtained by mating five sires from population A and 20 dams from population B from generation 10 of their respective populations. Subsequently, 20 females from the first hybrid generation were mated with five males from population B of generation 10. The two generations in the hybridization process were referred to as generations 11 and 12, respectively. Animals in the purebred reproduction breeding process were mated with their contemporaries, and five males and 20 females were selected to maintain the population in each generation. Five generations of the purebred reproduction breeding process were simulated to examine the effectiveness of diverse breeding methods.

#### 2.2.3. Evaluation Criteria

An animal’s true breeding value was used as an evaluation criterion to evaluate the effectiveness of different breeding methods. The advantageous genotype, as well as, the effect of each QTL was considered in the simulation, therefore, an individual’s true breeding value was calculated as follows:$$TBV_{i}=\sum_{j=1}^{nQTL}Z_{ij}a_{QTL_{j}},$$
where $TBV_{i}$ is the true breeding value of individual i, $a_{QTL}$ is the vector that includes the additive QTL effect values, and Z is the genotype matrix for all animals, which is coded by 0, 1, 2. All simulations were repeated 10 times and the resultant data were tested for significance using ANOVA.

### 2.3. Feasibility Test

Four selection methods, including KIS, GBLUP, BayesB, and strategy based on true breeding value (TBV), were implemented in the simulation. The effectiveness of each selection method was evaluated by the true breeding values of the simulated individual. In addition, negative control simulation was implemented by replacing the ideal individual with an individual with 600 noneffective loci and random beneficial alleles.

### 2.4. Robustness Test

Robustness tests were implemented to fit better replicate the real situation, including QTL with different false negative rates, QTL with different false positive rates, QTL with different quantitative gradients, and the scale of foundation population and selection proportion, and simulation of dominant and epistatic effects.

#### 2.4.1. QTL with Different False Negative Rates

QTL for the ideal individual was sorted considering the smallest to greatest effect values, that is, 10%, 20%, and 30% of the QTL were randomly selected for deletion to reconstruct the ideal individual (false negative individual), and the missing QTL of the ideal individual with a low absence rate was contained in the one with high absence rate. The selection and mating of simulated individuals were optimized according to the false negative individual. Nevertheless, the TBV of the mating individuals was calculated using true QTL effects.

The advanced KIS method did not consider QTL effects when establishing the ideal individual, therefore, we built an ideal individual containing beneficial genotypes and QTL effects. Other basic settings were as previously described.

#### 2.4.2. QTLs with Different Pseudo-Positive Rates

The pseudo-positive rates were the same as those in the false negative test, which included three gradients. However, pseudo-positive QTLs were adjacent to true QTLs on chromosomes. The false positive QTL did not duplicate true QTL, and beneficial genotypes were assigned randomly. Other basic settings were as previously described.

#### 2.4.3. QTLs with Different Quantitative Gradients

In addition to the combination of 100 and 500, which represent the number of QTLs for two quantitative traits, three other combinations, including, 100–350, 150–400, and 200–500 were considered to investigate the robustness of the KIS method. The number of QTLs for the other three selection methods subsequently changed. Other basic settings were as previously described.

#### 2.4.4. Scale of Foundation Population

In this context, different scales of the foundation population were taken into consideration to detect the robustness of the KIS method, while the number of QTLs was the same as in the previous simulation (100–500). Five different gradients of the foundation population sizes, that is, 2000, 1000, 500, 200, and 50, were utilized to examine if the KIS method was still valid in the disparate scale of the foundation population. The ratio of male to female was half in all the selection methods. Five sires and 20 dams were selected for mating according to the criteria corresponding to different selection methods. Other basic settings were as previously described.

#### 2.4.5. Selection Proportion

For the examination of the selection proportion, the scale of the foundation population was fixed at 200, with an equal number of sires and dams. Five selection proportions: 0.2–0.5, 0.15–0.4, 0.1–0.3, 0.05–0.2, and 0.03–0.1 were simulated. In these combinations, each group of figures represented the proportion of male and female animals in the foundation population. Other basic settings were the same as previously described.

#### 2.4.6. Simulation of Dominance and Epistatic Effects

In the simulation of dominance and epistatic effects, the impact of each effect on the accuracy of the KIS method was first simulated separately, followed by a simultaneous consideration of both effects on the simulation results. Different QTL gradient settings were employed in the simulation, representing 10%, 20%, and 30% of the total number of QTLs with additive effects.

## 3. Result

### 3.1. Negative Control Simulation

Figure 3 shows the comparison of the KIS method and negative control for determining the breeding values of boars in all generations. The comparison reflected that the breeding value of the boars in the KIS method increased in succeeding generations. Moreover, the breeding values of boars corresponding to the negative control remained unchanged except for in the F1 generation. This may be because the F1 boars were mainly heterozygotes, and heterozygotes were assigned a value of 1 during calculation. The same phenomenon was consistent in the selected group of sows.

### 3.2. Feasibility Test

The comparison of the KIS method with conventional breeding methods for determining the breeding values of boars and sows is illustrated in Figure 3. The breeding values corresponding to all breeding methods increased in succeeding generations. The TBV method had the highest breeding values among all generations, followed by the KIS method, whose value was higher than those of the BayesB and GBLUP methods. The results of the significance tests for breeding values are presented Table 1. The significance test results show no significant difference between the breeding values corresponding to the various breeding methods until the G3 generation (generation 3). In contrast, G4 generation (generation 4) onwards, there was a considerable difference between TBV and other breeding methods.

### 3.3. QTLs with Different False Negative Rates

The breeding values corresponding to the KIS method at different deletion rates are shown in Figure 4A,B. The breeding values corresponding to the KIS method increased in succeeding generations at different absence rates, and there was no apparent regularity in the scale of breeding values between absence rates. The significance test results are presented in Table 2; the differences in breeding values corresponding to absence deletion rates were not significant in any generation. 

Data corresponding to the KIS method with zero absence rate were selected for comparison with general breeding methods; the results were the same as those shown in Figure 3. Figure 4C,D shows the results of the comparison of four different absence rates with QTL effects taken into consideration. The results of the significance test (Table 2) indicate that there was no significant difference between the breeding values associated with varying rates of absence until generation G3, while the breeding values related to a 30% absence rate in generation G4 were significantly different. The breeding values associated with a 20% absence rate in generation G5 also show significant differences compared to the other absence rates.

### 3.4. QTLs with Different Pseudo-Positive Rates

The results of the pseudo-positive test are shown in Figure 5A,B. The comparison of breeding values for different pseudo-positive rates shows that the breeding value decreased as the pseudo-positive rate increased in succeeding generations. The significance test shows that the differences between breeding values corresponding to different pseudo-positive rates were not significant in any generation. The results of the significance tests for breeding values are shown in Table 3.

The result of the comparison of the KIS method with the other selection methods is presented in Figure 5C,D. In addition, data on breeding values relating to the KIS method with a 30% false positive rate were compared with those of the conventional methods and showed that the KIS method was less effective than the TBV method but better than the rest of the selection methods. The results of the significance tests for breeding values are presented in Table 3. The significance test results showed no significant difference between the breeding values corresponding to the various breeding methods until the G3 generation. In contrast, G4 generation onwards, there was a considerable difference between TBV and other breeding methods.

### 3.5. QTLs with Different Quantitative Gradients

The results of the three quantitative gradients regarding the number of different QTL tests are presented in Figure 6, and the significance test results are shown in Table 4. The results of breeding values show the same regularity regardless of the quantitative gradient, i.e., the KIS method is less effective than the TBV method but better than the other methods. Furthermore, the results of the significance test showed that the TBV method showed significant differences compared with the other selection methods only at generation G5 when the number of QTLs was small. In contrast, the TBV method showed significant differences compared with the other selection methods at generation G1 as the number of QTLs increased. The KIS method showed no substantial compared differences with the BayesB and GBLUP methods in any generation. 

### 3.6. Scale of Foundation Population

Figure 7A,B illustrates the results of the foundation population size simulation. The comparison reflected that breeding values increased as the scale of the foundation population increased, however, the difference in breeding values between the foundation population scales was not significant. Only at a foundation population size of 100 in the G5 generation, the breeding value was significantly different from those of the other foundation population scales. The results of the significance tests for breeding values with scales of different foundation population is shown in Table 5.

### 3.7. Selection Proportion

The selection proportion simulation experiment results are presented in Figure 7C,D and show that the breeding value is essentially inversely proportional to the selection proportion in both males and females. In addition, the significance test results showed that the effect of different selection proportions on breeding values does not differ significantly in all generations. The results of the significance tests for breeding values with different selection proportions are shown in Table 5.

### 3.8. Simulation Results of Dominance and Epistatic Effects

Figure 8 presents the results of considering only dominant effects, only epistatic effects, and both dominant and epistatic effects, from top to bottom, and the number of QTL from 10% to 30%, from left to right. Notably, when accounting for both dominant and epistatic effects, a declining trend in breeding values was observed for each generation as the number of QTL increased.

Compared to the conventional breeding method, individuals selected through the KIS method exhibited superior performance in terms of true breeding values, with overall higher breeding values than those selected by the conventional method, albeit with a slight reduction in just a few generations. Importantly, individuals selected based solely on true breeding values achieved the highest scores, and all KIS scores were inferior to these. In addition, the results of the significance test show that there is no significant difference in breeding value among KIS method and conventional breeding methods. The results of the significance tests for breeding values with different selection proportions are shown in Appendix A.

## 4. Discussion

In this study, we aimed to assess the effectiveness and stability of the KIS method in a breeding program. The KIS method is distinct from previous selection methods in the following ways: (1) The selection criteria are established by identifying an ideal individual, (2) The measurement of phenotypic traits is not a crucial aspect of the breeding process, and (3) A large reference population is not necessary, making the KIS method appropriate for small pig populations.

The effectiveness of genomic selection is dependent on the linkage disequilibrium between causative mutations and the SNP markers (generally numbering approximately 50,000) utilized in genomic predictions [16]. The dairy cattle industry has widely adopted this technology, with over 4 million animals genotyped via SNP arrays. Despite its success in narrowly defined populations, such as Holstein-Friesian dairy cattle [9]. However, in the field of pig breeding, the use of whole-genome markers is not feasible due to the multitude of indigenous pig breeds in China. Furthermore, the SNP markers predominantly utilized are derived from breeds such as Duroc, Landrace, and Large White, which exhibit a significant genetic distance from the indigenous pig breeds of China [17].

Under almost all the mentioned simulation conditions above, the selection effectiveness of the KIS method was better than that of the GBLUP and BayesB methods but lower than that of the TBV method across all generations. This may be due to the limited sample size in the simulation population, which hinders the full potential of the GBLUP and Bayes methods and reduces their accuracy, this also demonstrates that the KIS method has its own unique features in the case of small population size.

In the negative control simulation experiment, the breeding value of F1 was significantly lower than that at other ages. This may be because the F1 generation mainly included heterozygotes, and the heterozygotes were assigned a value of 1 during calculation. 

In the simulation experiment for false negatives, the performance of individuals selected through the equally weighted KIS method showed no decrease in their true breeding values as the rate of absence increased. However, this result was not in line with expectations as the individuals selected by the equally weighted KIS method were observed to have a higher abundance of favorable gene loci, however, the total of these loci did not yield the highest values in terms of breeding values, in the scenario of low absence rates. This discrepancy could also be due to a truncated selection mating strategy, as the breeding values of offspring produced through different mating combinations are not always positively correlated with the breeding values of their parents. When assigning equal weights, certain QTLs with large effects may not be favored solely based on the number of favorable genes they possess. However, by removing part of QTLs with minor effects, the weights of these large-effect QTLs are amplified, which could help explain this phenomenon. In other words, removing part of the influence of small-effect QTLs through the equal-weighted approach may enable the large-effect QTLs to receive the recognition they deserve. 

The simulation results for false positives were as anticipated, with a decrease in breeding values as the false positive rate increased. Despite this, the KIS method was generally consistent with both the GBLUP and BayesB methods, even when the false positive rate reached 30%.

The patterns observed in the three distinct QTL quantitative gradients were not completely congruent with prior simulations. While the patterns of the breeding values remained unchanged, the results of the significance tests revealed that as the number of QTLs increased, the TBV method demonstrated significant disparities compared to the other methods at earlier generations. For instance, in the first QTL quantitative gradient (350–100), significant differences were only noted in generation G5, while in the third QTL quantitative gradient (500–200), significant disparities appeared as early as generation G1. This may be attributed to the increase in the breeding value of the ideal individual with an increasing number of simulated QTL and the consequent loss of more QTL loci during the transmission hybridization process.

In practice, balancing breeding effectiveness with production costs requires controlling the scale of the foundation population within acceptable limits. Results from simulation studies indicated that increasing the size of the foundation population leads to a corresponding increase in breeding effectiveness. However, statistical significance tests of the KIS method revealed little difference between different foundation population scales in generations other than the fifth, where a foundation population size of 100 led to a substantial difference. Additionally, the results of the seed reservation simulations were as expected, with breeding values rising with each generation as reservation rates decreased. These results suggest that breeders using the KIS method have the flexibility to tailor the reservation rate and foundation population scale to meet specific requirement.

When dominant and epistatic effects were considered in the simulation, the selection effect of KIS gradually decreased as the proportion of dominant and epistatic effects QTL number increased, probably because the proportion of additive effects gradually decreased when dominant and epistatic effects were taken into account in the design of ideal individuals.

When considering only additive effects, we calculated the coefficient of variation corresponding to four different methods within the same generation, and the results are shown in the Appendix A. In the table, the coefficient of variation for the KIS method is very close to that of the TBV method. After the G2 generation, the CV for the KIS method is slightly higher than that of the TBV method, which may be due to the fact that the KIS method did not assign weights to each locus during selection.

The simulation study presented in this paper showcases the effectiveness of the KIS method in selecting individuals for breeding. Our model selectively bred individuals from the same generation, under similar environmental conditions, and with a focus on a few key traits. This method has proven to be effective in enhancing genetic gain while minimizing the risk of negative traits being unintentionally selected for. However, it is important to note that our model only considered individuals from the same generation and under similar environmental conditions, without accounting for external factors such as field effects and maternal effects. In the process of hybrid breeding, selection is carried out in the same generation and the same gender, considering only the genotypes and excluding the influence of environmental effects.

## 5. Limitations and the Future of Applying the KIS Method

The effectiveness of the kinship index based selection (KIS) method is limited by the accuracy and coverage of functional genes in pigs. Previous studies have reported on the association of functional genes with traits in pigs, and databases, such as The Farm animal Genotype-Tissue Expression TWAS server (http://twas.farmgtex.org/) [18] and PigVar [19], exist which combine non-redundant variations and evolutionary selective scores. Results from the simulation study indicated that the mating method used may impact selection by the KIS method and that the truncated selection method used may not be the most optimal. Previous research has suggested that incorporating advantageous effects into the mating method can balance genetic progression and genetic diversity, albeit with some loss of genetic progression [20,21,22]. In the KIS method, a truncation selection approach is adopted. After the selection of males and females is completed, they are randomly paired without considering the impact of inbreeding coefficients on breeding results. In actual breeding processes, conventional mating methods can be used to avoid rapid increases in inbreeding coefficients, some mating methods have been mentioned in the previous text. In subsequent studies of the KIS method, we will continue to research mating methods for improvement. In future studies, the authors aim to provide a database of beneficial genotypes and mating methods to support the KIS method.

## 6. Conclusions

The KIS method solves the problem of phenotypic determination and establishment of genomic reference populations in the breeding process by establishing an ideal individual; it is a breeding method that is easy to implement, has a wide range of applications, and has high accuracy. The application value of the KIS method will continue to improve as research on pig functional genes progresses.

## Figures and Tables

**Figure 1 genes-14-00807-f001:**
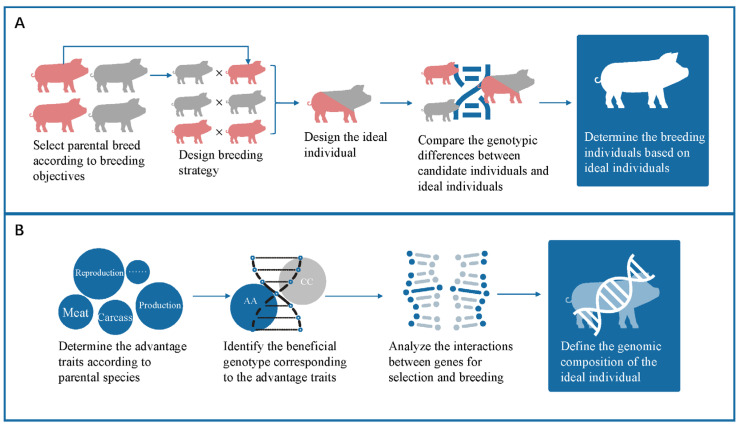
(**A**) The entire progress of the KIS method; (**B**) The progress of constructing the ideal individual.

**Figure 2 genes-14-00807-f002:**
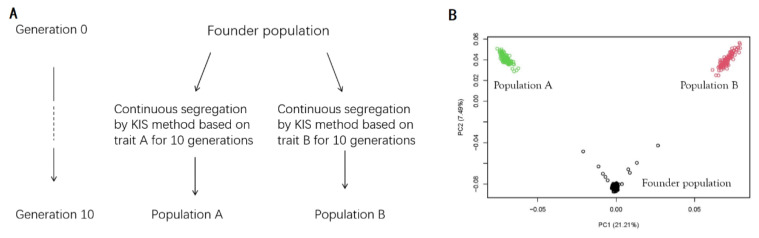
(**A**) The entire progress of simulating the historical population and (**B**) the principal com-ponent analysis of individuals in the founder population (black circle) and population A (green circle) and population B (red circle) of generation 10.

**Figure 3 genes-14-00807-f003:**
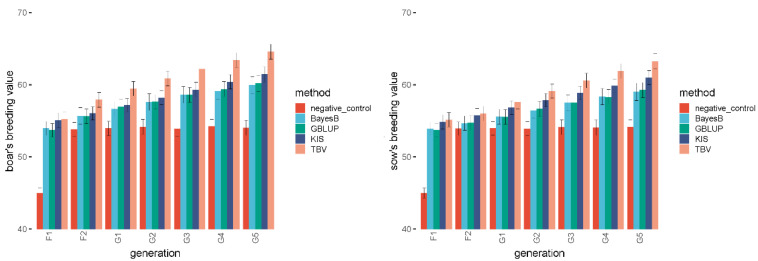
The comparison of the KIS method with negative control and general breeding methods for breeding values of boars and sows.

**Figure 4 genes-14-00807-f004:**
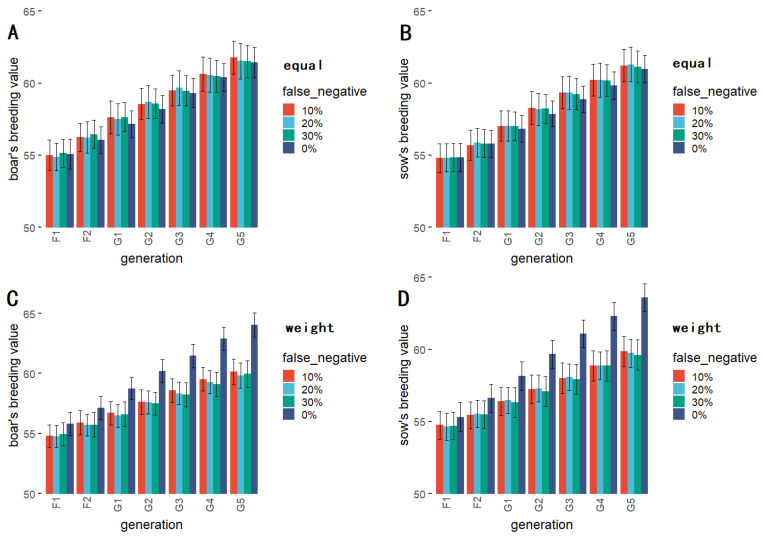
The results of comparing four different absence rates for equal-weight KIS method for boars (**A**) and sows (**B**); The results of comparing four different absence rates with QTL effects taken into consideration for weighted KIS method for boars (**C**) and sows (**D**).

**Figure 5 genes-14-00807-f005:**
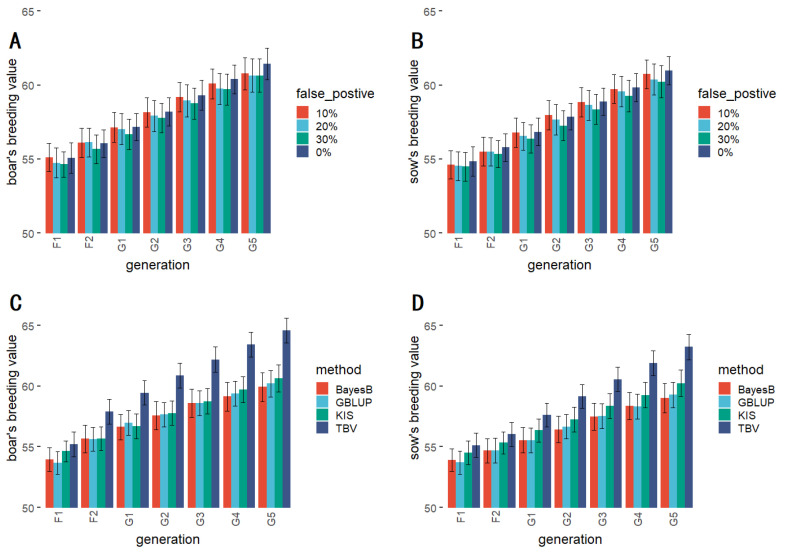
The comparison of breeding values for different pseudo-positive rates for boars (**A**) and sows (**B**); The comparison of the KIS method with general breeding methods for breeding values for boars (**C**) and sows (**D**).

**Figure 6 genes-14-00807-f006:**
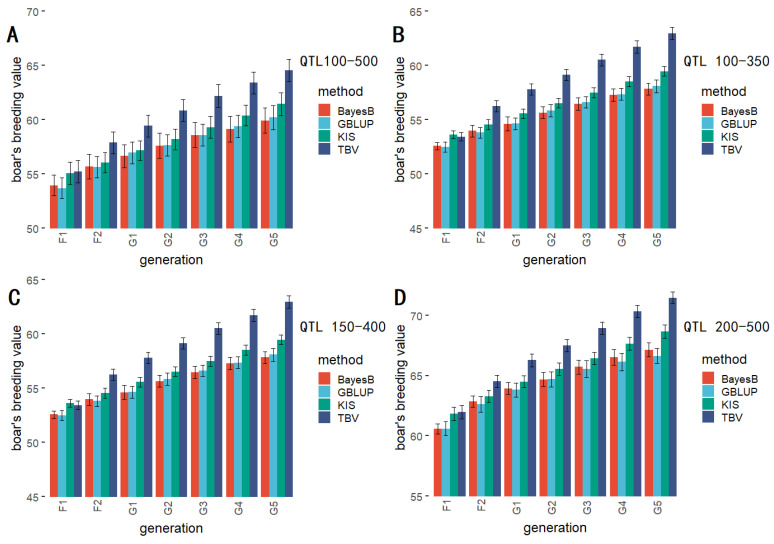
The comparison of the KIS method with general breeding methods for breeding values in different QTL quantitative gradients.

**Figure 7 genes-14-00807-f007:**
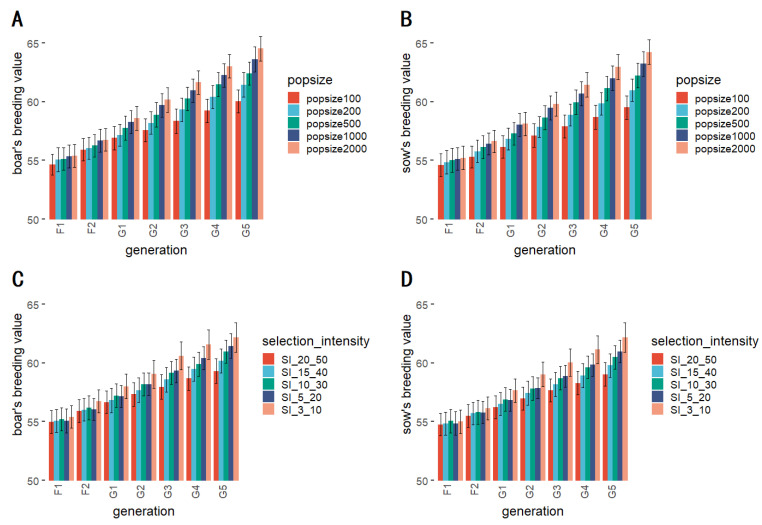
The breeding values corresponding to the KIS method at different scales of foundation population for boars (**A**) and sows (**B**); The breeding values corresponding to the KIS method at different selection proportions for boars (**C**) and sows (**D**).

**Figure 8 genes-14-00807-f008:**
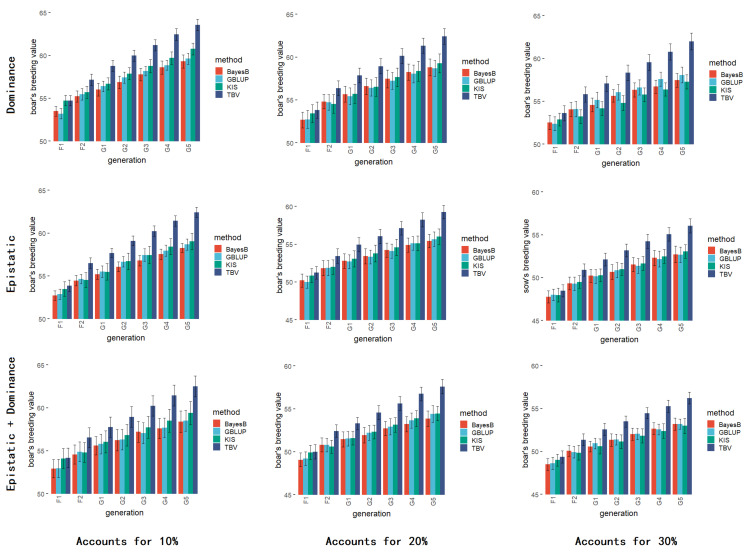
The result of simulation for non-additive effects. In the figure, from top to bottom, the addition of dominant effects, the addition of supernumerary effects, and the addition of both dominant and supernumerary effects; from left to right, the proportion of the number of non-additive effect QTL to the number of additive effect QTL, from 10% to 30%.

**Table 1 genes-14-00807-t001:** The results of the significance tests for breeding values. The same letter (A,B) used in the labeling of different methods indicates no significant difference, while different letters indicate a significant difference between the two methods.

Generation	F1	F2	G1	G2	G3	G4	G5
Method							
KIS	55.068 B	56.049 A	57.165 B	58.196 B	59.318 B	60.396 B	61.446 B
Negative control	45.003 A	53.809 A	53.958 A	54.179 A	53.873 A	54.262 A	54.011 A
GBLUP	53.685 A	55.624 A	56.949 A	57.648 A	58.585 A	59.396 B	60.199 B
BayesB	53.952 A	55.665 A	56.638 A	57.594 A	58.583 A	59.136 B	59.930 B
KIS	55.068 A	56.049 A	57.165 A	58.196 A	59.318 A	60.396 AB	61.446 AB
TBV	55.207 A	57.896 A	59.439 A	60.858 A	62.190 A	63.410 A	64.576 A

**Table 2 genes-14-00807-t002:** The results of the significance tests for breeding values in false negative simulation. The same letter used in the labeling of different methods indicates no significant difference, while different letters indicate a significant difference between the two methods.

Generation	F1	F2	G1	G2	G3	G4	G5
False Negative Rates							
0% (equal)	55.068 A	56.049 A	57.165 A	58.196 A	59.318 A	60.396 A	61.446 A
10% (equal)	55.014 A	56.242 A	57.620 A	58.552 A	59.498 A	60.633 A	61.768 A
20% (equal)	54.887 A	56.222 A	57.497 A	58.702 A	59.665 A	60.559 A	61.536 A
30% (equal)	55.137 A	56.462 A	57.636 A	58.582 A	59.462 A	60.473 A	61.499 A
0% (weight)	55.807 A	57.131 A	58.755 A	60.199 A	61.466 A	62.903 A	64.040 A
10% (weight)	54.789 A	55.891 A	56.724 A	57.622 A	58.589 A	59.534 AB	60.148 AB
20% (weight)	54.765 A	55.707 A	56.476 A	57.594 A	58.342 A	59.278 AB	59.843 B
30% (weight)	54.936 A	55.739 A	56.601 A	57.490 A	58.252 A	59.106 B	59.954 B

**Table 3 genes-14-00807-t003:** The results of the significance tests for breeding values in the pseudo-positive simulation. The same letter used in the labeling of different simulations indicates no significant difference, while different letters indicate a significant difference between the two simulations.

Generation	F1	F2	G1	G2	G3	G4	G5
Pseudo-Positive Test							
0%	55.068 A	56.049 A	57.165 A	58.196 A	59.318 A	60.396 A	61.446 A
10%	55.109 A	56.088 A	57.146 A	58.155 A	59.193 A	60.087 A	60.780 A
20%	54.733 A	56.124 A	57.032 A	57.928 A	58.946 A	59.749 A	60.649 A
30%	54.641 A	55.666 A	56.673 A	57.777 A	58.757 A	59.703 A	60.646 A
GBLUP	53.685 A	55.624 A	56.949 A	57.648 A	58.585 A	59.396 B	60.199 B
BayesB	53.952 A	55.665 A	56.638 A	57.594 A	58.583 A	59.136 B	59.930 B
KIS (30%)	54.641 A	55.666 A	56.673 A	57.777 A	58.757 A	59.703 B	60.646 B
TBV	55.207 A	57.896 A	59.439 A	60.858 A	62.190 A	63.410 A	64.576 A

**Table 4 genes-14-00807-t004:** The results of the significance tests for breeding values with QTL quantitative gradients. The same letter used in the labeling of different simulations indicates no significant difference, while different letters indicate a significant difference between the two simulations.

Generation	F1	F2	G1	G2	G3	G4	G5
QTL Quantitative Gradients							
GBLUP (QTL 100-350)	48.941 A	51.324 A	52.115 A	53.249 A	54.355 A	55.065 A	56.156 AB
BayesB (QTL 100-350)	49.161 A	51.138 A	52.353 A	53.312 A	53.979 A	54.609 A	55.551 B
KIS (QTL 100-350)	50.320 A	51.661 A	52.970 A	54.012 A	55.304 A	56.204 A	57.185 AB
TBV (QTL 100-350)	50.372 A	53.380 A	54.931 A	56.339 A	57.617 A	58.834 A	59.977 A
GBLUP (QTL 150-400)	52.4935 A	53.8096 B	54.6248 B	55.8352 B	56.4513 B	57.2635 B	58.1057 B
BayesB (QTL 150-400)	52.5697 A	53.9613 B	54.6381 B	55.6529 B	56.6019 B	57.3517 B	57.8415 B
KIS (QTL 150-400)	53.6179 A	54.5528 AB	55.5732 B	56.5278 B	57.4771 B	58.5359 B	59.4759 B
TBV (QTL 150-400)	53.4380 A	56.2417 A	57.8186 A	59.1561 A	60.5100 A	61.7221 A	62.9629 A
GBLUP (QTL 200-500)	60.5938 A	62.6374 A	63.8198 B	64.7000 B	65.5406 B	66.1291 B	66.6235 B
BayesB (QTL 200-500)	60.5819 A	62.8641 A	63.9442 B	64.6765 B	65.7173 B	66.5116 B	67.1417 B
KIS (QTL 200-500)	61.8339 A	63.2821 A	64.4985 AB	65.5540 AB	66.4141 B	67.6554 B	68.6712 B
TBV (QTL 200-500)	61.9914 A	64.5347 A	66.2707 A	67.4954 A	68.9448 A	70.3415 A	71.4575 A

**Table 5 genes-14-00807-t005:** The results of the significance tests for breeding values with different selection proportions.The same letter used in the labeling of different simulations indicates no significant difference, while different letters indicate a significant difference between the two simulations.

Generation	F1	F2	G1	G2	G3	G4	G5
Simulation Test							
Popsize 2000	55.391 A	56.751 A	58.575 A	60.159 A	61.646 A	63.016 A	64.532 A
Popsize 1000	55.363 A	56.690 A	58.272 A	59.692 A	60.943 A	62.250 A	63.591 AB
Popsize 500	55.130 A	56.260 A	57.748 A	58.893 A	60.252 A	61.464 A	62.400 AB
Popsize 200	55.068 A	56.049 A	57.165 A	58.196 A	59.318 A	60.396 A	61.446 AB
Popsize 100	54.641 A	55.911 A	56.907 A	57.583 A	58.345 A	59.235 A	60.039 B
SI_3_10	55.411 A	56.738 A	57.999 A	59.044 A	60.615 A	61.564 A	62.152 A
SI_5_20	55.068 A	56.049 A	57.165 A	58.196 A	59.318 A	60.396 A	61.446 A
SI_10_30	55.219 A	56.186 A	57.222 A	58.179 A	59.157 A	59.483 A	60.948 A
SI_15_40	55.051 A	56.009 A	56.810 A	57.689 A	58.574 A	59.483 A	60.184 A
SI_20_50	54.978 A	55.899 A	56.666 A	57.329 A	57.968 A	58.675 A	59.303 A

## Data Availability

All simulation-related codes are accessible at the following URLs: http://alphaindex.zju.edu.cn/ALPHADB/download.html.

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
