# Peer review of "Molecular Design-Based Breeding: A Kinship Index-Based Selection Method for Complex Traits in Small Livestock Populations"

_genes, 2023, doi:10.3390/genes14040807_

Round 1

Reviewer 1 Report

This is a very interesting manuscript and is presented very clearly. This research presents an alternative approach to inform breeding value of indigenous pig breeds that don't have the structure of the more prominant breeds. The idea is to look at the genetic similarity between animals and an ideal animal with all of the desirable identified alleles. 

I was fascinated by this work and I think it has a lot of potential value. I dont have anything to add to this work.

Author Response

March 21, 2023

Genes Journal

Dear Reviewer,

Thank you for your thoughtful and constructive review of our work. We appreciate your positive feedback, and we are thrilled to hear that our research has been well received. Your positive feedback is particularly encouraging, and it is gratifying to know that our research has resonated with you.

We hope you will find that our contribution is suitable for publication in Genes Journal, and we are looking forward to your comments.

Sincerely yours,

Qishan Wang, Ph.D.

Department of animal breeding and genetics

Zhejiang University

E-mail: wangqishan@zju.edu

Reviewer 2 Report

The method description and result presentation of this manuscript still need to be greatly improved. I suggest a revised review. The main comments are as follows.

(1)              The simulated data used in this paper is difficult to model for nonlinear data or polynomial regression with the correlation between data features. (2) It is difficult to express highly complex data well, so it is unscientific to verify in large-scale groups without using real data.

(2)              When selecting, various external factors, such as field effects, parent effects, etc., are not considered, and the construction of the model should be reconsidered.

(3)              The difficulty of making the selection is to first figure out the degree of correlation between two individuals, the genealogy of a breeding farm, usually more than 3 generations, if bred to ten generations, the complexity between the pedigree will gradually increase, more and more complex, how to avoid inbreeding in small-scale groups when doing simulation.

(4)              From the overall effect of the comparison, the error rate of KIS is low, but TBV is higher than the estimated breeding value of traditional methods (GBLUP and BayesB). Please explain in more detail in the discussion

(5)              What is the theoretical basis for designing quantitative gradients for different QTLS? Is there a golden scale theory to support it?

Above all, a lot of grammatical mistakes have been observed, especially in the introduction, the results and the discussion. There is a need to rule out these deficiencies.

Author Response

March 21, 2023

Genes Journal

Dear Reviewer,

The method description and result presentation of this manuscript still need to be greatly improved. I suggest a revised review. The main comments are as follows.

Response: I would like to express my sincere appreciation for your constructive feedback and thoughtful review of our work. Your comments have been incredibly helpful in improving the quality of our manuscript, and we are grateful for the time and effort you have invested in reviewing our work. Thank you again for your invaluable contributions to our work. We look forward to continuing our engagement with you and other members of the research community as we work to further refine and develop our research.

  1. The simulated data used in this paper is difficult to model for nonlinear data or polynomial regression with the correlation between data features. (2) It is difficult to express highly complex data well, so it is unscientific to verify in large-scale groups without using real data.

Response: Thank you for your thorough review of our work and for your insightful feedback.

We use the MoBPS R package to simulate the effects of dominance and epistasis on our study. In the case of considering only additive effects, the selection effect of the KIS method is better than that of the GBLUP and BayesB methods. After adding dominance and epistasis effects to the model, the selection effect of the KIS method may be slightly lower than that of the GBLUP or BayesB methods in some generations, but the Single Factor Analysis of Variance (ANOVA) showed no significant differences in all generations. However, the KIS method still has the advantage of not requiring phenotype determination. The results of the significance tests for breeding values with different selection proportions are shown in supplementary file. We found that as the number of QTLs associated with dominance and epistasis increased, the accuracy of the KIS method decreased. This is because the proportion of additive effects in the idealized individual decreases when accounting for the effects of dominance and epistasis.

We have supplemented the content related to this part in the methods, results, and discussion sections. Please refer to lines 215-220, 325-344, and 378-384 of the article.

Figure 8. In the figure, from top to bottom, the addition of dominant effects, the addition of epistatic effects, and the addition of both dominant and epistatic effects; from left to right, the proportion of the number of non-additive effect QTL to the number of additive effect QTL, from 10% to 30%.

Supplementary Table1. The results of the significance tests for breeding values with different selection proportions.

genenation

F1

F2

G1

G2

G3

G4

G5

simulation test

GBLUP(dominance 10%)

53.485 A

55.4695 A

56.3978 A

57.4165 A

58.1771 A

58.8652 A

59.6308 A

BayeB(dominance 10%)

53.2072 A

55.2232 A

56.0266 A

56.8639 A

57.768 A

58.5975 A

59.3002 A

KIS(dominance 10%)

54.7107 A

55.6954 A

56.7003 A

57.8833 A

58.7541 A

59.7001 A

60.7548 A

TBV(dominance 10%)

54.7301 A

57.1591 A

58.7729 B

59.9683 B

61.1921 B

62.4555 B

63.5511 B

GBLUP(dominance 20%)

52.731 A

54.725 A

55.428 A

56.423 A

57.233 A

58.057 A

58.685 A

BayeB(dominance 20%)

52.66 A

54.818 A

55.679 A

56.598 A

57.462 A

58.247 A

58.799 A

KIS(dominance 20%)

53.418 A

54.534 A

55.721 A

56.534 A

57.667 A

58.363 A

59.286 A

TBV(dominance 20%)

53.816 A

56.375 A

57.867 A

58.939 A

60.139 B

61.336 B

62.424 B

GBLUP(dominance 30%)

52.368 A

54.091 A

55.165 A

56.056 A

56.62 A

57.587 A

58.08 A

BayeB(dominance 30%)

52.534 A

54.045 A

54.582 A

55.631 A

56.335 A

56.709 A

57.462 A

KIS(dominance 30%)

52.876 A

53.209 A

54.157 A

54.789 A

55.767 A

56.383 A

57.276 A

TBV(dominance 30%)

53.607 A

55.81 A

57.068 A

58.328 B

59.543 B

60.763 B

61.98 B

GBLUP(epistatic 10%)

52.8763 A

54.6505 A

55.509 A

56.6655 A

57.452 A

57.9583 A

58.6803 A

BayeB(epistatic 10%)

52.7231 A

54.4565 A

55.198 A

56.0743 A

56.79 A

57.5413 A

58.2693 A

KIS(epistatic 10%)

53.4724 A

54.5096 A

55.471 A

56.7153 A

57.443 A

58.4309 A

59.0313 A

TBV(epistatic 10%)

53.8563 A

56.5087 A

57.687 A

59.0997 B

60.233 B

61.4288 B

62.4329 B

GBLUP(epistatic 20%)

50.004 A

51.862 A

52.707 A

53.328 A

54.082 A

55.101 A

55.662 A

BayeB(epistatic 20%)

50.2 A

51.824 A

52.785 A

53.413 A

54.214 A

54.891 A

55.427 A

KIS(epistatic 20%)

50.843 A

51.982 A

53.063 A

53.784 A

54.59 A

55.131 A

56.005 A

TBV(epistatic 20%)

51.23 A

53.443 A

54.953 A

56.034 A

57.122 A

58.223 A

59.256 B

GBLUP(epistatic 30%)

47.978 A

49.3 A

50.138 A

50.873 A

51.379 A

52.177 A

52.66 A

BayeB(epistatic 30%)

47.759 A

49.338 A

50.213 A

50.676 A

51.533 A

52.307 A

52.722 A

KIS(epistatic 30%)

47.77 A

49.499 A

50.289 A

50.989 A

51.663 A

52.485 A

53.06 A

TBV(epistatic 30%)

48.45 A

50.907 A

52.125 A

53.191 A

54.254 A

55.062 A

56.055 B

GBLUP(dominance + epistatic 10%)

52.956 A

54.881 A

55.794 A

56.311 A

57.032 A

57.666 A

58.476 A

BayeB(dominance + epistatic 10%)

52.909 A

54.566 A

55.588 A

56.232 A

57.198 A

57.588 A

58.375 A

KIS(dominance + epistatic 10%)

54.109 A

54.805 A

56.029 A

56.819 A

57.715 A

58.477 A

59.383 A

TBV(dominance + epistatic 10%)

54.168 A

56.528 A

57.737 A

58.942 A

60.21 A

61.431 A

62.51 A

GBLUP(dominance + epistatic 20%)

49.202 A

50.79 A

51.52 A

52.192 A

52.909 A

53.64 A

54.392 A

BayeB(dominance + epistatic 20%)

49.058 A

50.802 A

51.432 A

51.937 A

52.7 A

53.212 A

53.847 A

KIS(dominance + epistatic 20%)

49.913 A

50.578 A

51.58 A

52.297 A

53.126 A

53.871 A

54.456 A

TBV(dominance + epistatic 20%)

49.989 A

52.409 A

53.286 A

54.572 A

55.632 A

56.739 B

57.601 B

GBLUP(dominance + epistatic 30%)

48.6207 A

49.91 A

50.9243 A

51.38 A

52.0317 A

52.564 A

53.174 A

BayeB(dominance + epistatic 30%)

48.4844 A

50.064 A

50.5589 A

51.3231 A

51.9912 A

52.645 A

53.183 A

KIS(dominance + epistatic 30%)

48.9964 A

49.854 A

50.5981 A

51.1282 A

51.8052 A

52.372 A

52.973 A

TBV(dominance + epistatic 30%)

49.3494 A

51.348 A

52.5579 A

53.4821 A

54.439 A

55.263 B

56.22 B

  1. When selecting, various external factors, such as field effects, parent effects, etc., are not considered, and the construction of the model should be reconsidered.

Response: Thank you for point this out., we have added new content to the discussion section to address the points you raised.

It is important to note that our model only considered individuals from the same generation and under similar environmental conditions, without accounting for external factors such as field effects and maternal effects. In the process of hybrid breeding, selection is carried out in the same generation and the same gender, considering only the genotypes and excluding the influence of environmental effects. Please refer to lines 419-428 of the article.

  1. The difficulty of making the selection is to first figure out the degree of correlation between two individuals, the genealogy of a breeding farm, usually more than 3 generations, if bred to ten generations, the complexity between the pedigree will gradually increase, more and more complex, how to avoid inbreeding in small-scale groups when doing simulation.

Response: Thank you for point this out. The KIS method utilizes a truncation selection approach, where males and females are selected and randomly paired without considering inbreeding coefficients. To avoid rapid increases in inbreeding coefficients, conventional mating methods can be used in actual breeding processes. In subsequent studies, we will investigate additional mating methods to further improve the KIS method. Our future goal is to develop a database of beneficial genotypes and mating methods to support the KIS method. Please refer to lines 434-441 of the article.

  1. From the overall effect of the comparison, the error rate of KIS is low, but TBV is higher than the estimated breeding value of traditional methods (GBLUP and BayesB). Please explain in more detail in the discussion

Response: Thank you for your remind. When considering only additive effects, we calculated the coefficient of variation corresponding to four different methods within the same generation, and the results are shown in the table below. In the table, the coefficient of variation for the KIS method is very close to that of the TBV method. After the G2 generation, the CV for the KIS method is slightly higher than that of the TBV method, which may be due to the fact that the KIS method did not assign weights to each locus during selection. Please refer to lines 419-424 of the article.

Supplementary Table2. The coefficients of variation for the four methods when only additive effects are considered

Generation

F1

F2

G1

G2

G3

G4

G5

Method

GBLUP

0.055

0.057

0.057

0.054

0.055

0.055

0.058

BayesB

0.057

0.065

0.059

0.064

0.063

0.063

0.062

KIS

0.058

0.056

0.051

0.052

0.054

0.051

0.055

TBV

0.06

0.055

0.053

0.053

0.053

0.051

0.05

  1. What is the theoretical basis for designing quantitative gradients for different QTLS? Is there a golden scale theory to support it?

Response: Thank you for your helpful comments. The design of the QTL quantity was to match the characteristic of low heritability traits having more QTLs in real-life situations. In this paper, the purpose is to simulate whether the performance of the KIS method remains stable under different genetic power gradients corresponding to different numbers of QTL. Three different gradients of QTL numbers were simulated separately to compare the performance of the KIS method with three other methods in breeding values. The gradients of QTL numbers are shown in the table below.

QTL number

QTL number

QTL number

Trait 1

350

400

500

Trait 2

100

150

200

Sincerely yours,

Qishan Wang, Ph.D.

Department of animal breeding and genetics

Zhejiang University

E-mail: wangqishan@zju.edu

Round 2

Reviewer 2 Report

To Editor Genes

Note of Acceptance for article entitled “Molecular design-based breeding: a kinship index-based selection method for complex traits in small livestock populations.

It seems that the authors have provided a professional response to my previous comments.

Regarding my comments, the authors have provided additional information to address the queries. They have explained how they used the MoBPS R package to simulate the effects of dominance and epistasis and provided results showing the selection effects of their method compared to other methods. They also explained that the KIS method still has the advantage of not requiring phenotype determination.

 The authors have supplemented the content related to the section I mentioned and provided a figure and a supplementary table to support their claims. They also included the specific lines in the manuscript where readers can find the additional information.

Overall, it seems that the authors have taken my comments seriously and made appropriate revisions to address my concerns. Although the revised manuscript is sufficient for publication but still, I found a myriad of grammatical errors throughout the entire manuscript, warranting substantial improvements. It is with great sincerity that I urge the authors to consider enhancing the manuscript's eloquence and precision, as it holds great potential to contribute immensely to the scientific community.